# Innate Immune Response of *TmToll-3* Following Systemic Microbial Infection in *Tenebrio molitor*

**DOI:** 10.3390/ijms24076751

**Published:** 2023-04-04

**Authors:** Maryam Ali Mohammadie Kojour, Ho Am Jang, Yong Seok Lee, Yong Hun Jo, Yeon Soo Han

**Affiliations:** 1Department of Applied Biology, Institute of Environmentally-Friendly Agriculture (IEFA), College of Agriculture and Life Sciences, Chonnam National University, Gwangju 61186, Republic of Korea; 2Department of Biology, College of Natural Sciences, Soonchunhyang University, Asan 31538, Republic of Korea

**Keywords:** *Toll-3*, *Tenebrio molitor*, microbial infection, RNAi, antimicrobial peptides, AMP assay

## Abstract

Although Toll-like receptors have been widely identified and functionally characterized in mammalian models and *Drosophila*, the immunological function of these receptors in other insects remains unclear. Here, we explored the relevant innate immune response of *Tenebrio molitor* (*T. molitor*) *Toll-3* against Gram-negative bacteria, Gram-positive bacteria, and fungal infections. Our findings indicated that *TmToll-3* expression was mainly induced by *Candida albicans* infections in the fat bodies, gut, Malpighian tubules, and hemolymph of young *T*. *molitor* larvae. Surprisingly, *Escherichia coli* systemic infection caused mortality after *TmToll-3* knockdown via RNA interference (RNAi) injection, which was not observed in the control group. Further analyses indicated that in the absence of *TmToll-3*, the final effector of the Toll signaling pathway, antimicrobial peptide (AMP) genes and relevant transcription factors were significantly downregulated after *E. coli* challenge. Our results indicated that the expression of almost all AMP genes was suppressed in silenced individuals, whereas the expression of relevant genes was positively regulated after fungal injection. Therefore, this study revealed the immunological involvement of *TmToll-3* in *T. molitor* in response to systematic infections.

## 1. Introduction

Toll and Toll-like receptors (TLRs) play a major role in the innate immunity of insects and mammals [1]. Toll, which was the first of these receptors to be discovered, was initially noted for its role in dorsal–ventral patterning in *Drosophila* embryos [2] and was later shown to participate in the immune response against fungi and Gram-positive bacteria in larvae and adults [3]. Much of what is known about Toll receptors in insects derives from studies on *Drosophila melanogaster*, and a total of nine Toll receptors (Toll, Toll-2 also known as 18-Wheeler, and Toll-3 to Toll-9) have thus far been identified in this model [4]. Except for the *Toll-3* (also known as *MstProx*) and *Toll-4* genes, which have no clear functional or phenotypic characteristics [5], most of the *Drosophila* Toll paralogs (Toll-1, -2, -5, -6, -7, -8, also known as Tollo, and -9) have been found to play important roles in early development and the immune response [6,7,8,9,10,11,12]. Toll-2, along with Toll-1, plays a key role as an adhesion molecule in *Drosophila* compartment boundaries [13,14] and salivary gland morphogenesis [15]. Encoding more leucine-rich repeats (LRRs) is a shared feature that is believed to give Toll-2, -6, -7, and -8 rigid interaction and common roles in *Drosophila* embryonic elongation [16]. In addition to its role in development, 18-Wheeler is reported to have antibacterial activity in *Drosophila* larvae [17]. Previous studies have reported that Toll-5 induces the mRNA expression of the gene encoding for anti-microbial peptides (AMPs) such as drosomycin and metchnikowin [18]. Moreover, Toll-6 and Toll-7 participate in several functions within the *Drosophila* nervous system, including the wiring of neural circuitry, removal of apoptotic debris from neurons [16], motor axon targeting, neuronal survival [19], and axon and dendrite targeting [20]. Among the wide variety of biological processes in which the Toll signaling pathway is reportedly involved, Toll-7 has been reported to regulate tumor growth and invasion [21]. Moreover, Toll-7 has been reported to participate in antiviral defense [12]. Among the remaining Toll receptors that have been identified thus far in *Drosophila*, Toll-9 indirectly mediates the production of AMPs, which led to the hypothesis that it might be involved in innate immunity [10]. However, loss-of-function experiments failed to confirm this hypothesis [22], suggesting that Toll-9 may exhibit functional redundancy with other Toll proteins [22]. Other studies have revealed that Toll-9 strongly induces all hallmarks of undead apoptosis-induced proliferation (AiP) signaling, including Duox-dependent reactive oxygen species [17] generation, hemocyte recruitment, and JNK signaling [23]. A study that characterized the functions of the Toll-9 receptor in the silk moth *Bombyx mori* revealed that *Bm*Toll-9 had immunological roles through the recognition of lipopolysaccharide (LPS) [24]. Identification studies of TLRs in crustaceans further reported that Toll-9 in *Penaeus monodon* could induce signaling cascades downstream of the Toll pathway, leading to the activation of NF-κB transcription response elements [25].

The domain structure of Toll/Toll-like receptors is conserved across animal phyla, resulting in a realm of similarities and differences among different species [26]. Overall, the structure of Toll receptors and their signaling pathway in insects share basic characteristics and compartments. Regarding receptor structure, Tolls have an extracellular leucine-rich repeat (LRR) domain, a single-pass transmembrane domain, and a cytoplasmic Toll/Interleukin-1 receptor (TIR) domain [27,28]. Although all mammalian TLRs are known to play immunological roles [29], four Toll genes (Toll-1, -5, -7, and -9) have also been found to participate in immune responses [5].

A total of seven Toll genes (Toll-2, -3, -6, -7, -8, -9, and -10) have been identified in the yellow mealworm beetle (*Tenebrio molitor*) [21,30]. Unlike mammalian TLRs, *T. molitor* TLRs are not directly triggered by structurally conserved molecules derived from microbes or other possible stimuli, but are instead activated by the cytokine ligand Spätzle (Spz) [26]. We have identified 9 Spz genes in *T. molitor* [31,32,33,34,35], which are cleaved following a proteolytic cascade, leading to the binding of the mature active C-terminal C-106 domain of Spz to the Toll receptor [36]. Following the recognition of the Lys-type peptidoglycan (PGN) of the cell wall of Gram-positive bacteria by PGN recognition protein (PGRP-SA)/Gram-negative binding protein 1 (GNBP1), β-1,3-glucans of yeast and some fungi by GNBP3 (Appendix A), and (unlike *Drosophila*) polymeric DAP-type PGN of some Gram-negative bacteria, the downstream *Tenebrio* modular serine protease (ModSP) activates a cascade of CLIP-domain zymogens comprising the serine protease Spätzle-processing enzyme (SPE)-activating enzyme (SAE) and SPE [37,38,39].

Several studies have recently investigated the immunological response of Toll receptors in *T. molitor* following systemic infection. Moreover, although most studies on the immunological functions of insects have largely focused on dipterans [40], the importance of mealworm (*T. molitor*) rearing in various industrial fields and their nutritional value have recently garnered increasing attention, including the molecular mechanisms that drive their innate immune response, and the pathways involved in these responses, particularly Toll signaling [26]. Our research group had previously reported that *T. molitor* Toll-7 and Toll-2 exhibited anti-bacterial activity against *Escherichia coli* (*E. coli*) infection [30,37]. Here, we examine the role of the Toll-like receptor 3 gene from *T. molitor* (*TmToll-3*) in vivo through gene knockout experiments using RNAi technology. Consistent with previous studies on *TmTolls*, our findings reveal that *TmToll-3* plays a key role in mediating the immune response against Gram-negative bacteria such as *E. coli*.

## 2. Results

### 2.1. Sequence Analysis of TmToll-3

In this study, a Toll-3 homolog from *T. molitor* (*TmToll-3*, Accession number: OP566500) was identified through an expressed sequence tag (EST) and RNA-seq search using the *T. castaneum* protein sequence as a query. Phylogenetic analysis based on the full-length amino acid sequences of *TmToll-3* and other insect Toll receptors indicated that it clustered closely with the Toll-3 proteins from *Drosophila melanogaster* toll, isoform C, *Aedes aegypti* protein toll, *Lucilia sericata* protein toll-like isoform, *Bm*TollX2 (*Bombyx mori* protein toll isoform X2), *Gm*Toll-like (*Galleria mellonella* protein toll-like), *Manduca sexta* protein toll, *Nylanderia fulva* protein toll, *Homalodisca vitripennis* toll-like receptor 7, and *Mus musculus* toll-like receptor 2 isoform X1 (Figure 1). Furthermore, within this branch, *TmToll-3* appears to be most closely related to *Tc*Toll (31% aa identity) from *T. castaneum*, which belongs to the same insect order as *T. molitor* (Coleoptera).

### 2.2. Expression Analysis of TmToll-3 Transcripts

We next investigated the expression pattern of *TmToll-3* by qRT-PCR at different developmental stages (egg, young-instar larvae, late-instar larvae, prepupae, 1- to 7-day-old pupae, and 1- to 5-day-old adults). Our results demonstrated that the *TmToll-3* expression level was high in the embryonic stage but then gradually decreased between the pre-pupal and late-pupal stages (Figure 2A). We further examined the transcript levels of *TmToll-3* in different tissues of late-instar larvae and adults. As shown in Figure 2B, *TmToll-3* expression was relatively high in the integument of late-instar larvae and low in the fat bodies and Malpighian tubules. In contrast, in adult tissues, *TmToll-3* was predominantly expressed in the gut and to a lesser extent in other tissues, including fat bodies, ovaries, and testes (Figure 2C). Unlike in the larvae, the lowest expression of *TmToll-3* was detected in the integument of *T. molitor* adults.

Next, to determine whether *TmToll-3* expression was regulated in response to immune challenge, we further examined the temporal changes in *TmToll-3* mRNA expression in *T. molitor* larvae after infection with either Gram-negative (*E. coli*), or Gram-positive (*S. aureus*) bacteria, or a fungus (*C. albicans*). In brief, total RNA was isolated from control and immune-challenged larvae (10th or 11th instar) at 3, 6, 9, 12, and 24 h post-infection, followed by reverse-transcription and qRT-PCR using *TmToll-3*- specific primers. As shown in Figure 3A–D, *TmToll-3* was induced after challenging the larvae with *S. aureus* and *C. albicans*. However, in the case of *E. coli* infection, except for Malpighian tubules in which the *TmToll-3* mRNA levels were marginally increased within 6 to 12 h post-infection, relevant expression in other tissues was insignificant compared with the PBS-injected larvae.

### 2.3. T. molitor Larval Mortality Assay

Given that our findings indicated that *TmToll-3* expression was induced after infection with *S. aureus*, *C. albicans*, and *E. coli*, we next sought to determine whether *TmToll-3* played a role in the immune response against bacteria and fungi by monitoring the survival rates of infected *T. molitor* larvae after treatment with either control dsRNA (*TmVer*) or *TmToll-3* dsRNA. As illustrated in Figure 4A, *TmToll-3* mRNA levels decreased by 75% in larvae 7 days after injection with *TmToll-3* dsRNA compared with those treated with control dsRNA. After confirming the efficient knockdown of *TmToll-3*, we then challenged the ds*TmToll-3*-treated and control larvae by injecting them with 1 µL of bacterial (*E. coli* or *S. aureus*, 1 × 10^6^/µL) or fungal suspension (*C. albicans*, 5 × 10^4^/µL) and monitored their survival for 10 days. Surprisingly, the ds*TmToll-3* larvae were significantly more susceptible to *E. coli* infection (36 vs. 13% mortality) compared with the ds*TmVer* (control) larvae, (Figure 4B), whereas their survival rates after infection with *S. aureus* or *C. albicans* were not affected compared with the controls (Figure 4C,D).

### 2.4. Effects of TmToll-3 RNAi on the Expression of AMP and Other Downstream Signaling Genes in Response to Microorganism Infection

Considering the contrasting findings related to the induction of *TmToll-3* and the survival of *T. molitor* larvae after ds*TmToll-3* treatment, we next sought to determine whether *TmToll-3* mediated any of the observed effects through AMP production. Therefore, *TmToll-3* was once again knocked out and the expression levels of 15 different AMP genes were measured following challenge with *E. coli*, *S. aureus*, and *C. albicans*. The aim of this experiment was to identify AMPs that were significantly induced upon infection by *E. coli* but reversed by *TmToll-3* knockdown. In turn, this would suggest that *TmToll-3* at least partially mediates the activation of these AMPs in response to *E. coli* infection. Among the 15 AMPs tested, except for *TmTenecin-2* and *TmThaumatin like protein-1*, the mRNA expression levels of all the AMPs were induced in response to *E. coli* infection, but pretreatment with *TmToll-3* dsRNA suppressed their upregulation (Figure 5A–O, Appendix A). Interestingly, *TmToll-3* knockdown also suppressed the mRNA levels of *TmTenecin-1*, *TmTenecin-4*, *TmDefensin-like*, *TmCecropin*, *TmColeoptericin-A*, *TmAttacin-1a*, *TmAttacin-1b*, and *TmAttacin-2* in *S. aureus*-challenged larvae. Finally, none of the 15 AMPs showed significant responses to *C. albicans* infection, regardless of whether *TmToll-3* was knocked down prior to infection.

To further examine the role of *TmToll-3* in regulating the immune response against pathogens, we investigated how *TmToll3* RNAi might affect the expression of Toll pathway-related transcription factor genes (*TmDorsal1* and *TmDorsal-2*) and one Imd-related gene (*TmRelish*) using the specific primers listed in Table 1. Our results demonstrated that the expression of the Toll pathway-related genes was reduced by *TmToll-3* RNAi (*p* < 0.05) following *E. coli* infection, whereas the level of *TmRelish* was upregulated (Figure 6). This suggests that *TmToll-3* can positively regulate genes downstream of the Toll signaling pathway and that *TmToll-3* functions through the Toll signaling pathway to regulate AMP expression.

## 3. Discussion

Our assessment of the expression of AMPs in *T. molitor* in response to *TmToll-3* knockdown and pathogen challenge provides key insights into the mechanisms that govern insect innate immunity. Upstream of this response is the recognition of microbial particles by special pattern recognition receptors (PRRs) that trigger proteolytic cascades, leading to the activation of signaling pathways responsible for AMP production [41,42]. While exploring the immunological roles of *TmToll-3* in *T. molitor*, we found that Toll receptors and their possible ligands, 9 Spzs isoforms, reduced the induction of several AMPs [31,32,33,34,35], which we found to be suppressed by *TmToll-3* RNAi after *E. coli* challenge. A previous study reported that out of the 9570 putative orthologs of annotated *T. castaneum* genes, 213 were related to immune functions in infected *T. molitor* [43]. While determining which of these receptors were involved in AMP production, we demonstrated that *Tm*Toll2 and *Tm*Toll7 participated in the immune response by regulating specific NF-kB transcription factors through preliminary knockdown studies [30,37].

The dynamic expression pattern of Toll proteins has also been reported in *Drosophila* [44]. Our results indicated that the mRNA of *TmToll-3* was not only expressed during the embryonic stage but also in 5-day-old pupae, suggesting that Tolls play important roles throughout the insect’s lifespan. Furthermore, unlike other Tolls in non-infected *T. molitor*, the highest *TmToll-3* expression levels were observed in the integument of larvae and the gut of adults in our tissue-specific gene expression experiments, suggesting that similar to the *Drosophila*, this protein is closely linked to cuticle formation during *T. molitor* metamorphosis [45]. It would thus be interesting to investigate the evolutionary and developmental crosstalk between chitin and Toll expression and distribution. Unlike other Tolls in *T. molitor*, the mRNA expression of *TmToll-3* in the most important immune organs of larvae exhibited low induction upon pathogen infection. However, similar to the expression of *TmToll-2* [30], the main response occurred following *C. albicans* injection.

Therefore, for the remainder of our study, we focused on the effects of *TmToll3* silencing in larvae. Unlike in *Drosophila*, where *E. coli* DAP-type peptidoglycans can only activate Imd signaling, purified PGRP-SA and GNBP1 (upstream of Toll receptors) mediate the cleavage of pro-Spätzle in *T. molitor* larvae, thus demonstrating that *E. coli* is able to activate Toll signaling in this insect [39,46]. Similarly, both PGN and LPS can induce the expression of AMP genes in Lepidoptera such as *Bombyx mori* and *Manduca sexta* [47]. Moreover, *Drosophila* might also be more sensitive to PGN than to LPS because it carries 13 genes that belong to the PGRP family and therefore expresses a wider repertoire of PGRP proteins [48]. Consistent with previous findings, the decreased survival rates of *TmToll-3*-silenced *T. molitor* larvae following *E. coli* infection observed herein suggested that humoral innate immunity could occur via *TmToll-3*. *TmToll3* silencing in larvae rendered larvae more susceptible to *E. coli* infection and suppressed certain AMP genes induced by *E. coli* challenge, including *TmTenesin-1, -4, TmDefensin, TmDefensin-like, TmColeoptericin-A, -B, -C*, and *TmAttacin-1a, -1b, -2*, all four of which belong to AMP families known to exhibit antibacterial activity against Gram-negative bacteria [49,50,51,52]. *Tm*Toll-7 and *Tm*Toll-2 have been reported to activate their downstream NF-kB transcription factor, which leads to the induction of immune response genes including AMP genes [30,37]. Other studies have reported orthologs of transcription factors in *T. molitor*, including two Dif orthologs and two Relish orthologs [43]. Additionally, recent studies have characterized the signal activation further upstream of *T. molitor* NF-kB transcription factors and their relevant final effectors following bacterial and fungal infection [53,54]. Based on our findings that *TmToll-3* RNAi inhibits AMP expression, we hypothesize that *TmToll-3* in *T. molitor* plays an important role in mediating nuclear translocation of transcription factors for activating AMP gene expression. Our findings demonstrated that, after *TmToll-3* activation in *T. molitor*, Dif, *Tm*Dorx2, and potentially a Dif-Relish heterodimer stimulate the production of AMP genes as the final effectors. Our findings also demonstrated the lack of AMP specificity after the activation of NF-kB transcription factors following infection with different pathogens.

Additional studies are thus needed to shed light on what makes *TmToll-3* different from other Tolls in *T. molitor* that mediate immune responses, as well as whether their distinctions and similarities stemmed from evolutionary convergence or another mechanism entirely. Additionally, it would be interesting to assess which of the nine identified Spz ligands interact directly with *Tm*Toll3 (or other elements) and whether their interactions are relevant to immunity and/or development in *T. molitor*.

## 4. Materials and Methods

### 4.1. Insect Rearing and Microbial Infection

*T. molitor* larvae were reared on a wheat bran diet at 27 ± 1 °C, a 60 ± 5% relative humidity, and under dark conditions. All experiments were conducted with 10–12th instar larvae. To investigate the immunological function of *TmToll-3* against infections, three microorganisms, including *E. coli* K12, *Staphylococcus aureus* RN4220, and *Candida albicans* were used. Overnight cultures of *E. coli*, *S. aureus*, and *C. albicans* were grown in Luria-Bertani (LB) broth and Sabouraud Dextrose broth at 37 °C, respectively. The microorganisms were harvested, washed, and suspended in phosphate-buffered saline (PBS, pH 7.0) by centrifugation at 3500 rpm for 10 min, and the concentrations were measured at OD600. Finally, 10^6^ cells/µL of *E. coli* and *S. aureus* and 5 × 10^4^ cells/µL of *C. albicans* were injected into the larvae.

### 4.2. In Silico Analysis of TmToll-3

To perform Local-tblastn analysis, specific gene sequence of *TmToll-3* (accession number: OP566500) was obtained from RNAseq analysis and NCBI Expressed Sequence Tag database and the *T. castaneum* TLR-3 amino acid sequence (accession number: EEZ99323.1) was used as a query. The full-length open reading frame (ORF) and deduced amino acid sequences of *Tm*Toll-3 were analyzed using BLASTp (NCBI; https://blast.ncbi.nlm.nih.gov/Blast.cgi). The multiple sequence alignment of the *Tm*Toll-3 amino acid sequence with representative TLR amino acid sequences from other insects (retrieved from GenBank) was generated using ClustalX 2.1 [55] and MEGA 6 programs [56] to estimation of the percent identity and phylogenetic analyses respectively. The phylogenetic tree was constructed based on amino acid sequence alignments via the maximum likelihood method [38] (bootstrap trial set to 1000) with several protein sequences, including those of *Tc*Toll, *Tribolium castaneum* protein Toll (XP_967796.2); *Dm*TollIsoformC, *Drosophila melanogaster* Toll, isoform C (NP_001262995.1); *Aa*Toll, *Aedes aegypti* protein Toll (XP_021708718.1); *Ls*Toll-likeX1, *Lucilia sericata* protein Toll-like isoform (X1 XP_037811182.1); *Bm*TollX2, *Bombyx mori* protein Toll isoform X2 (XP_037870104.1); *Gm*Toll-like, *Galleria mellonella* protein Toll-like (XP_031767681.1); *Ms*Toll, *Manduca sexta* protein Toll (XP_037303038.1); *Nf*Toll, *Nylanderia fulva* protein Toll (XP_029178173.1); and *Hv*TLR7, *Homalodisca vitripennis* Toll-like receptor 7 (XP_046670491.1). *Mm*TLR2X1 *Mus musculus* Toll-like receptor 2 isoform X1 (XP_006501523.1) amino acid sequences were used as an outgroup.

### 4.3. Expression and Induction Patterns of TmToll-3

The developmental and tissue-specific expression patterns of *TmToll-3* were investigated via real-time quantitative reverse transcription PCR (qRT-PCR) using an Exicycler Real-Time PCR Quantification System (Bioneer Co., Daejeon, South Korea). To investigate developmental and tissue-specific expression patterns of *TmToll-3*, samples were collected from various developmental stages including the late instar larval, pre-pupal, 1–7-day-old pupal, and 1–2-day-old adult stages, and tissues were dissected from late instar larvae (integument, gut, fat body, Malpighian tubules, and hemolymph) and 5-day old adult (integument, gut, fat body, Malpighian tubules, hemolymph, ovary, and testis) individuals. To examine the induction patterns of *TmToll-3* upon microorganism challenge, *E. coli* (10^6^ cells/µL), *S. aureus* (10^6^ cells/µL), and/or *C. albicans* (5 × 10^4^ cells/µL) were injected into *T. molitor* larvae and samples were collected at 3, 6, 9, 12, and 24 h post-injection. PBS-injected *T. molitor* larvae were used as a negative control.

RNA was isolated using the Clear-S Total RNA Extraction Kit (Invirustech Co., Gwangju, Republic of Korea) according to the manufacturer’s instructions. Next, 2 μg of total RNA was used as a template to synthesize cDNA via Oligo(dT)-primed synthesis [12,13,14,15,16,17,18] under the following reaction conditions: 72 °C for 5 min, 42 °C for 1 h, and 94 °C for 5 min. These procedures were conducted using a MyGenie96 Thermal Block (Bioneer, Daejeon, Korea) and AccuPower^®^ RT PreMix (Bioneer Co., Daejeon, South Korea) according to the manufacturer’s instructions. The cDNA was then stored at –20 °C until further use. To investigate the expression levels of *TmToll-3* transcripts, qRT-PCR was conducted using AccuPower^®^ 2X GreenStar qPCR Master Mix (Bioneer), using the synthesized cDNAs as templates and gene-specific primers (*TmToll-3*_qPCR_Fw and *TmToll-3*_qPCR_Rv) (Table 1) at an initial denaturation step at 94 °C for 2 min followed by 35 cycles of denaturation at 94 °C for 30 s, annealing at 53 °C for 30 s, and extension at 72 °C for 30 s, with a final extension step at 72 °C for 5 min. *T. molitor* ribosomal protein (*TmL27a*) was used as an internal control and relative gene expression was calculated via the 2^−ΔΔCt^ method.

### 4.4. RNA Interference Analysis

The PCR product (510 bp sequence) containing the T7 promoter sequences was amplified using the AccuPower^®^ Pfu PCR PreMix polymerase with *TmToll-3*_T7_Fw and Rv primers (Table 1) using the same PCR conditions described above. dsRNA for *TmToll-3* was synthesized using the AmpliScribe T7-Flash Transcription Kit (Epicentre, Madison, WI, USA) and was purified with PCI (Phenol:Chloroform:Isopropyl alcohol mixture), followed by ammonium acetate purification and ethanol precipitation. Finally, 2 µg of synthesized *dsTmToll-3* was injected into 10–11th instar larvae for gene silencing and *T. molitor* vermilion (*TmVer*) double-strand RNA (ds*TmVer*) was used as a control [57].

### 4.5. Effect of TmToll-3 Knockdown on the Response to Microorganisms

To investigate the effect of *TmToll-3* knockdown on the systemic response to microbial infection, 10^6^ cells/µL of *E. coli* and *S. aureus*, and 5 × 10^4^ cells/µL of *C. albicans* were injected into ds*TmToll-3*-treated *T. molitor* larvae, respectively. Larval mortality was monitored up to 10 days post-injection of microorganisms. Ten insects per group were used for this assay and the experiments were replicated three times.

### 4.6. Effect of TmToll-3 RNAi on AMP Expression in Response to Microorganisms

To characterize the function of *TmToll-3* on the humoral innate immune response, gene knockdown experiments were conducted through *TmToll-3* RNAi injection, after which the *T. molitor* larvae were infected with the microorganisms (*E. coli*, *S. aureus,* and *C. albicans*) via injection. Samples were then collected at 24 h post-infection. PBS was used as an injection control and ds*TmVer*-treated *T. molitor* was used as a negative control. qRT-PCR was then conducted to characterize the temporal expression patterns of 15 antimicrobial peptide (AMP) genes, including *Tm*Tene-1 (Figure 5A: *Tm*Tenecin-1), *Tm*Tene-2 (Figure 5B: *Tm*Tenecin-2), *Tm*Tene-3 (Figure 5C: *Tm*Tenecin-3), *Tm*Tene-4 (Figure 5D: *Tm*Tenecin-4), *Tm*Def (Figure 5E: *Tm*Defensin), *Tm*Def-like (Figure 5F: *Tm*Defensin-like), *Tm*Cec-2 (Figure 5G: *Tm*Cecropin-2), *Tm*Cole-A (Figure 5H: *Tm*Coleoptericin-A), *Tm*Cole-B (Figure 5I: *Tm*Coleoptericin-B), *Tm*Cole-C (Figure 5J: *Tm*Coleoptericin-C), *Tm*Att-1a (Figure 5K: *Tm*Attacin-1a), *Tm*Att-1b (Figure 5L: *Tm*Attacin-1b), *Tm*Att-2 (Figure 5M: *Tm*Attacin-2), *Tm*TLP-1 (Figure 5N: *Tm*Thaumatin-like protein-1), and *Tm*TLP-2 (Figure 5O: *Tm*Thaumatin-like protein-2). Table 1 summarizes the sequences of the gene-specific primers used in this study.

### 4.7. Statistical Analysis

All experiments were performed in triplicate and all data are presented as means ± standard error (SE). Differences between groups were evaluated via one-way analysis of variance (ANOVA) and Tukey’s multiple range tests. *p*-values < 0.05 were considered statistically significant.

## Figures and Tables

**Figure 1 ijms-24-06751-f001:**
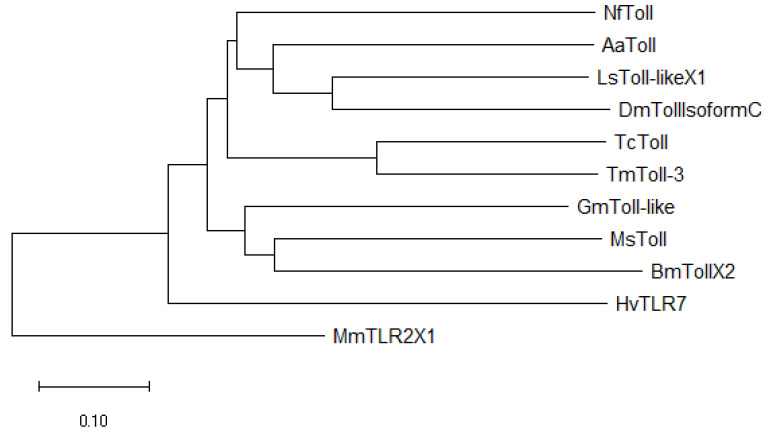
Molecular phylogenetic analysis of *TmToll-3* (*T. molitor Toll-3*) (OP566500). The phylogenetic tree was constructed using MEGA7 using the maximum likelihood method and 1000 bootstrap replicates (the numbers at the nodes indicate bootstrap support). The percentage of trees in which the associated taxa clustered together is shown next to the branches. The phylogenetic tree was constructed based on amino acid sequence alignments using several protein sequences, including those of *Tc*Toll, *Tribolium castaneum* protein Toll (XP_967796.2); *Dm*TollIsoformC, *Drosophila melanogaster* Toll isoform C (NP_001262995.1); *Aa*Toll, *Aedes aegypti* protein toll (XP_021708718.1); *Ls*Toll-likeX1, *Lucilia sericata* protein Toll-like isoform (X1 XP_037811182.1); *Bm*TollX2, *Bombyx mori* protein Toll isoform X2 (XP_037870104.1); *Gm*Toll-like, *Galleria mellonella* protein Toll-like (XP_031767681.1); *Ms*Toll, *Manduca sexta* protein Toll (XP_037303038.1); *Nf*Toll, *Nylanderia fulva* protein Toll (XP_029178173.1); and *Hv*TLR7, *Homalodisca vitripennis* Toll-like receptor 7 (XP_046670491.1). *Mm*TLR2X1 *Mus musculus* Toll-like receptor 2 isoform X1 (XP_006501523.1) amino acid sequences were used as an outgroup.

**Figure 2 ijms-24-06751-f002:**
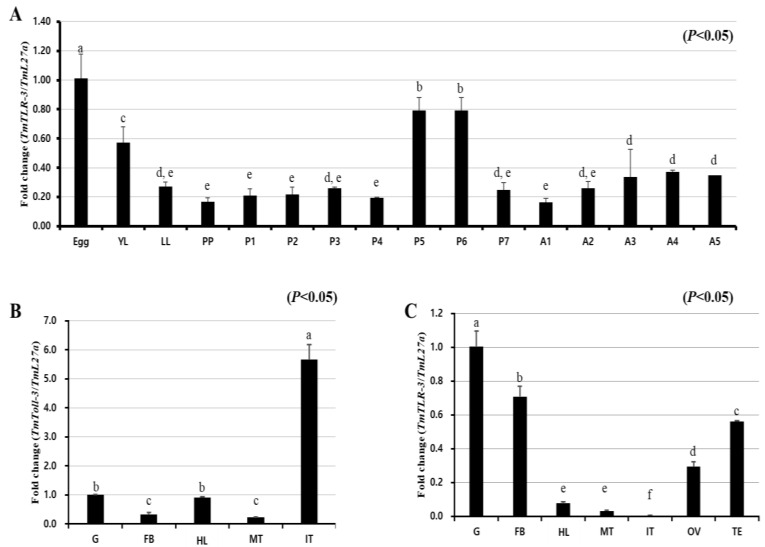
Developmental stage- and tissue-specific expression patterns of *TmToll-3* measured by qRT-PCR. (**A**) Relative mRNA expression levels of *TmToll-3* in eggs, young larvae, late-instar larvae, pre-pupae, 1- to 7-day-old pupae (P1–P7), and 1- to 5-day-old adults (A1–A5). The expression levels were highest in the eggs. The mRNA expression decreased at the larval stage and was lowest at the young larval stage. *TmToll-3* tissue expression patterns in late instar larvae (**B**) and adults (**C**) were also examined. Total RNA was extracted from different tissues, including the integument, Malpighian tubule (MT), gut (G), hemolymph (HL), and fat bodies (FB) of late instar larvae and the IT, MT, G, hemolymph, FB, ovary (OV), and testis (TE) of 5-day-old adults. Total RNA was isolated from 20 mealworms, and *T. molitor* 60S ribosomal protein 27a (*TmL27a*) primers were used as an internal control (*n* = 3). Comparisons between groups were made via one-way ANOVA and Tukey’s multiple-range test. Different letters above each bar indicate statistically significant differences according to Tukey’s multiple-range test (*p* < 0.05).

**Figure 3 ijms-24-06751-f003:**
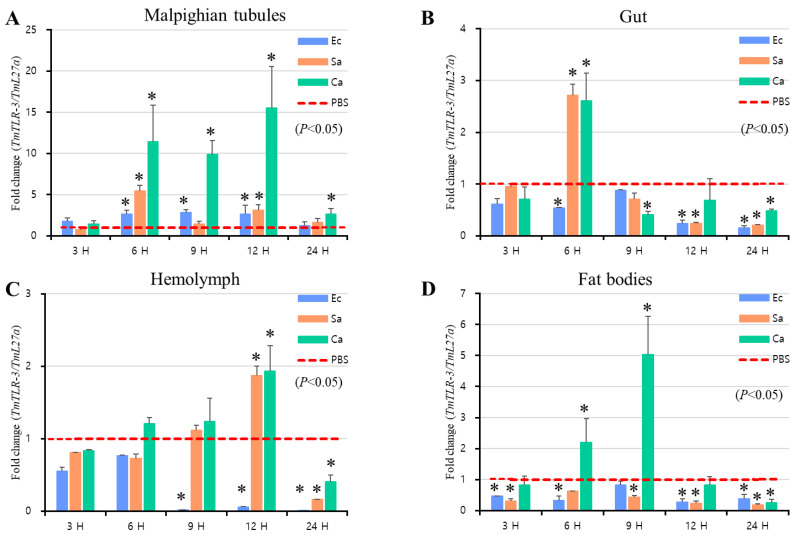
mRNA expression patterns of *TmToll-3* in immune-challenged *T. molitor* larvae. mRNA levels of *TmToll-3* in the Malpighian tubules (**A**), gut (**B**), hemolymph (**C**), and fat bodies (**D**) were examined by qRT-PCR 3, 6, 9, 12, and 24 h after infection with *E. coli* (10^6^ cells/µL), *S. aureus* (10^6^ cells/µL), and *C. albicans* (5 × 10^4^ cells/µL). *TmToll-3* mRNA expression was upregulated in response to all infectious sources and exhibited tissue- and time-dependent variations. The highest *TmToll-3* expression level was observed in the Malpighian tubules in response to *C. albicans* challenge. PBS was used as an injection control and *T. molitor* 60S ribosomal protein 27a (*TmL27a*) primers were used as an internal control to quantify relative gene expression (*n* = 3). The asterisks indicate significant differences between infected and PBS-injected larval groups as determined using Student’s *t*-test (*p* < 0.05). The vertical bars indicate means ± SD for each experimental condition (*n* = 20).

**Figure 4 ijms-24-06751-f004:**
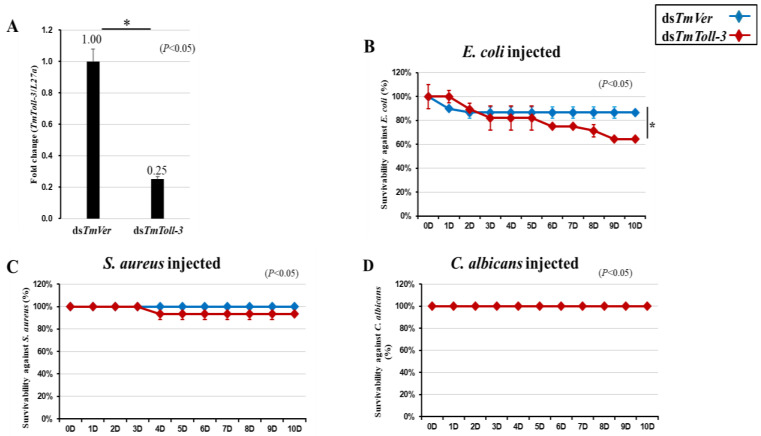
Effect of *TmToll-3* gene silencing on *T. molitor* larval survival. The RNAi efficiency of ds*TmToll-3* was measured by qRT-PCR 4 days after injection (**A**). *TmToll-3*-silenced larvae were injected with *E. coli* (**B**), *S. aureus* (**C**), and *C. albicans* (**D**), and survival rates were monitored for 10 days post-pathogen injection (*n* = 10 per group). The larval survival rates at 10 days post-microbial injection were 60% after *E. coli* injection and 100% after *S. aureus* and *C. albicans* injection compared with the survival rates of the ds*TmVer*-injected control group. Data were reported as averages of three biologically independent replicates. The asterisks indicate significant differences between the ds*TmToll-3*- and ds*TmVer*-injected groups. Survival analysis was performed using Kaplan–Meier plots (log-rank chi-squared test; * *p* < 0.05).

**Figure 5 ijms-24-06751-f005:**
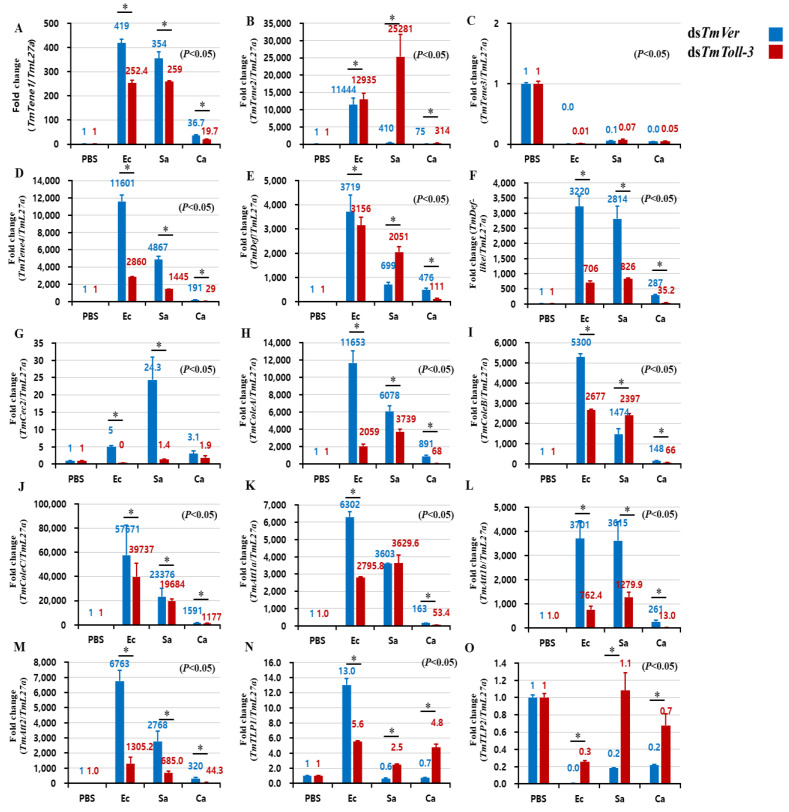
Induction of 15 AMP genes in the whole-body tissues of *TmToll-3*-treated *T. molitor* larvae infected with *E. coli* (Ec), *S. aureus* (Sa), and *C. albicans* (Ca) (PBS was used as a control). At 24 h post-infection, AMP genes including *TmTene1* (**A**), *TmTene2* (**B**), *TmTene3* (**C**), *TmTene4* (**D**), *TmDef* (**E**), *TmDef-like* (**F**), *TmCec2* (**G**), *TmColeA* (**H**), *TmColeB* (**I**), *TmColeC* (**J**), *TmAtt1a* (**K**), *TmAtt1b* (**L**), *TmAtt2* (**M**), *TmTLP1* (**N**), and *TmTLP2* (**O**) were examined via qRT-PCR using ds*TmVer* as a knockdown control and *T. molitor* ribosomal protein (*TmL27a*) as an internal control. All experiments were performed in triplicate. The asterisks indicate significant differences between the ds*TmToll-3*- and ds*TmVer*-treated groups, as determined by Student’s *t*-test (*p* < 0.05).

**Figure 6 ijms-24-06751-f006:**
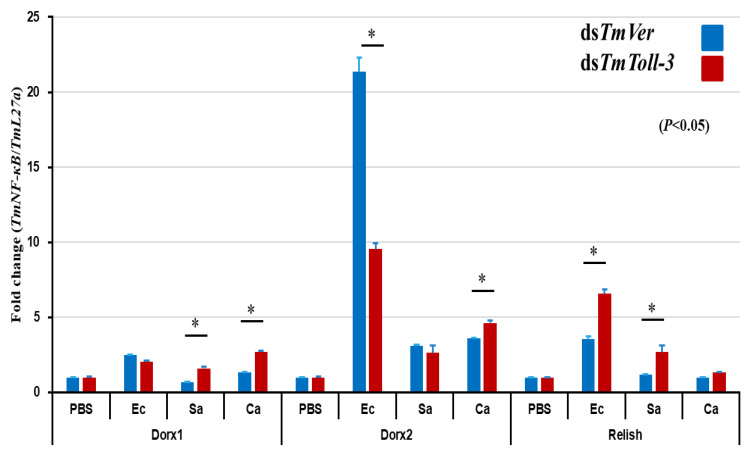
Effect of *TmToll-3* gene silencing on NF-kB gene expression. ds*TmToll-3*-treated *T. molitor* larvae were infected with *E. coli* (Ec), *S. aureus* (Sa), and *C. albicans* (Ca); at 24 h post-infection, whole-body mRNA levels of the NF-kB pathway genes *TmDorX1*, *TmDorX2*, and *TmRelish* were measured via RT-qPCR. The expression level of *TmDorx2* was suppressed following *E. coli* infection in the whole-body tissues, whereas the level of *TmRelish* was slightly positively regulated. *TmVer* dsRNA was assessed as a negative control and *T. molitor* ribosomal protein (*TmL27a*) was used as an internal control. All experiments were performed in triplicate. The asterisks indicate significant differences between ds*TmToll-3*- and ds*TmVer*-treated groups determined using Student’s *t*-test (*p* < 0.05).

**Table 1 ijms-24-06751-t001:** Sequences of the primers used in this study.

Name	Primer Sequences
*TmToll-3*_qPCR_Fw *TmToll-3*_qPCR_Rv	5′-GTTGGAGAATGTTGTCGGTG-3′ 5′-CGAACGATGTCGTCAATCTG-3′
*TmToll-3*_T7_Fw	5′-TAATACGACTCACTATAGGGT GACACGTTCATCAACAACGG-3′
*TmToll-3_*T7_Rv	5′-TAATACGACTCACTATAGGGT CGTTTTGGTTAAAGGCGAAA-3′
*dsTmVermillion*_Fw	5′-TAATACGACTCACTATAGGGT TCGAGAAGTCAGAGCAGCAA-3′
*dsTmVermillion*_Rv	5′-TAATACGACTCACTATAGGGT ACCACCAGTTCCCAGTTGAG-3′
*TmTenecin-1*_qPCR_Fw *TmTenecin-1*_qPCR_Rv	5′-CAGCTGAAGAAATCGAACAAGG-3′ 5′-CAGACCCTCTTTCCGTTACAGT-3′
*TmTenecin-2*_qPCR_Fw *TmTenecin-2*_qPCR_Rv	5′-CAGCAAAACGGAGGATGGTC-3′ 5′-CGTTGAAATCGTGATCTTGTCC-3′
*TmTenecin-3*_qPCR_Fw *TmTenecin-3*_qPCR_Rv	5′-GATTTGCTTGATTCTGGTGGTC-3′ 5′-CTGATGGCCTCCTAAATGTCC-3′
*TmTenecin-4*_qPCR_Fw *TmTenecin-4*_qPCR_Rv	5′-GGACATTGAAGATCCAGGAAAG-3′ 5′-CGGTGTTCCTTATGTAGAGCTG-3′
*TmDefensin*_qPCR_Fw *TmDefensin*_qPCR_Rv	5′-AAATCGAACAAGGCCAACAC-3′ 5′-GCAAATGCAGACCCTCTTTC-3′
*TmDefensin-like*_qPCR_Fw *TmDefensin-like*_qPCR_Rv	5′-GGGATGCCTCATGAAGATGTAG-3′ 5′-CCAATGCAAACACATTCGTC-3′
*TmColeoptericin-A*_qPCR_Fw *TmColeoptericin-A*_qPCR_Rv	5′-GGACAGAATGGTGGATGGTC-3′ 5′-CTCCAACATTCCAGGTAGGC-3′
*TmColeoptericin-B*_qPCR_Fw *TmColeoptericin-B*_qPCR_Rv	5′-CAGCTGTTGCCCACAAAGTG-3′ 5′-CTCAACGTTGGTCCTGGTGT-3′
*TmColeoptericin-C*_qPCR_Fw *TmColeoptericin-C*_qPCR_Rv	5′-GGACGGTTCTGATCTTCTTGAT -3′ 5′-CAGCTGTTTGTTTGTTCTCGTC-3′
*TmAttacin-1a*_qPCR_Fw *TmAttacin-1a*_qPCR_Rv	5′-AAAGTGGTCCCCACCGATTC-3′ 5′-GCGCTGAATGTTTTCGGCTT-3′
*TmAttacin-1b*_qPCR_Fw *TmAttacin-1b*_qPCR_Rv	5′-GAGCTGTGAATGCAGGACAA-3′ 5′-CCCTCTGATGAAACCTCCAA-3′
*TmCecropin-2*_qPCR_Fw *TmCecropin-2*_qPCR_Rv	5′-TACTAGCAGCGCCAAAACCT-3′ 5′-CTGGAACATTAGGCGGAGAA-3′
*TmDorsal-1*_qPCR_Fw *TmDorsal-1*_qPCR_Rv	5′-AGCGTTGAGGTTTCGGTATG-3′ 5′-TCTTTGGTGACGCAAGACAC-3′
*TmDorsal-2*_qPCR_Fw *TmDorsal-2*_qPCR_Rv	5′-ACACCCCCGAAATCACAAAC-3′ 5′-TTTCAGAGCGCCAGGTTTTG-3′
*TmRelish*_qPCR_Fw *TmRelish*_qPCR_Rv	5′-AGCGTCAAGTTGGAGCAGAT-3′ 5′-GTCCGGACCTCAAGTGT-3′
*TmL27a*_qPCR_Fw *TmL27a*_qPCR_Rv	5′-TCATCCTGAAGGCAAAGCTCCAGT-3′ 5′-AGGTTGGTTAGGCAGGCACCTTTA-3′

The underlined sequences indicate T7 promoter sequences.

## Data Availability

The authors confirm that the data supporting the findings of this study are available within the article Appendix A.

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
