# Peer review of "Innate Immune Response of TmToll-3 Following Systemic Microbial Infection in Tenebrio molitor"

_ijms, 2023, doi:10.3390/ijms24076751_

Round 1

Reviewer 1 Report

The study titled “Innate immune response of TmToll-3 following systemic microbial infection in Tenebrio molitor” by Maryam et al is interesting and novel. The whole study is based on qpcr and there are no further studies conducted to confirm the results of the study. The results of the study are not explained in detail and needs more elaboration. I have provided my suggestions and comments that the authors should consider mandatorily to improve the study.

1.      The results are a bit confusing. The mRNA expression of Toll-3 is upregulated when infected with C. albicans, however, when the gene is knocked out, the expression of AMPs is higher in E. coli. The authors need to justify how the whole mechanism is regulated and needs to provide a figure explaining a possible explanation of mechanism when exposed to bacteria and fungi. This will provide a better understanding for the readers.

2.      Figure 1: The labelling in the phylogenetic tree should be uniform and specific.

3.      Figure 2: “The expression levels were the highest in the eggs and 1 day old adults.” The same is not reflected in the figure. There is a discrepancy in the figure caption and the figure.

4.      Figure 1 caption: N should be given either in upper case and lower case throughout the manuscript.

5.      Figure 1 caption: The x axis labels in the figures do not correspond or are missing in the figure caption. For eg. Gut is represented as GT in caption and G in x-axis label.

6.      Figure 1B and 1C mRNA expression levels do not match that with figure 1A. For instance: expression levels in guts in 1B and 1C are many folds higher than shown for the stages shown in figure 1A.

7.      Figure 2 does not align with figure 1 data. Address the comment with respect to comment 6.

8.      Section 2.3: “Given that our findings indicated……” the statement is very misleading as the Toll3 expression was not induced in all the infections (figure 2). This needs to be justified.

9.      It is interesting that even after RNAi addition, there is around 0.25 fold expression observed. Has such a fold increase been recorded before?

10.   How were the survivability percentage calculated? The values from the graph does not look to be in whole number percentages since the number of individuals considered is a whole number of 10. It will be good to mention the formula used for calculation.

11.   The discussion is very short and does not discuss the results. The authors need to explain how the results are related between sections and with other data.

12.   Page 8: Line 242: “their possible ligands, spzs isoforms….” This is not shown as a results in the study?

13.   Page 8: line 256: “cuticle formation during insect metamorphosis..” mention that the results were observed for Drosophila and not T. molitor.

14.   Table 1: mention the annealing temperatures for each primer used for the qpcr study.

15.   Table 1: mention symbol mentioned at the end of the table at the appropriate primer names.

16.   Figure 5 can be shown as a single heatmap rather than representing it in so many graphs.

17.   Page 2: Line 81: “Several studies…” cite reference.

18.   There are few typographical errors and scientific misrepresentations in the manuscript. This needs to be proof-read and corrected.

19.   Since the study is completely based on qpcr results, it will be good and interesting to show the significant results of the study in terms of protein expression. A western blot with fold change protein expression can be given for the major results to prove the hypothesis in the study.

20.   Provide all qpcr generated graphs as supplementary file.

Author Response

Rebuttal letter to reviewer’s comments on our submitted manuscript (2294615)

We received the reviewers’ comments for the submitted manuscript (2294615). We are sincerely grateful for the reviewers’ positive and constructive comments, which have been very helpful in improving the quality of our paper. The suggestions have been noted, and we have worked on the reviewer comments. We have taken care of all comments and manuscript has been edited based on the comments. We hope the manuscript has improved with the suggested revision and would be a novel statement to the scientific community covered by the journal.

Please find the author’s comments to reviewers queries as under:

Independent Review Report:

Reviewer #1 (Comments to the Author): 

The study titled “Innate immune response of TmToll-3 following systemic microbial infection in Tenebrio molitor” by Maryam et al is interesting and novel. The whole study is based on qpcr and there are no further studies conducted to confirm the results of the study. The results of the study are not explained in detail and needs more elaboration. I have provided my suggestions and comments that the authors should consider mandatorily to improve the study.

  1. The results are a bit confusing. The mRNA expression of Toll-3 is upregulated when infected with C. albicans, however, when the gene is knocked out, the expression of AMPs is higher in E. coli. The authors need to justify how the whole mechanism is regulated and needs to provide a figure explaining a possible explanation of mechanism when exposed to bacteria and fungi. This will provide a better understanding for the readers.

Author’s response: We are deeply grateful for your constructive and fine comments. Kindly mark that as it has been described within introduction and discussion sections, seven TLR has been identified in T. molitor and contradictory results following the systemic infection in different time points and different distributions of different TLR in these tissue in the presence and absence of TmToll-3 can be also related to gene redundancy and homeostasis in T. molitor related to other isoforms of this gene or cross regulation with other signaling pathways involve in innate immunity. However, please note that according to your suggestion we have prepared a figure (Supplementary) as an explanation of activating mechanisms to simplify our study for the readers.

  1. Figure 1: The labelling in the phylogenetic tree should be uniform and specific.

Author’s response: Thank you for this fine notice. However, this is a very report over identification of one the TLR isoforms, therefore, illustrating a tree based on the amino acid sequence alignments using several protein sequences of different taxa it was not available to find identical sequences with similar labels. And the labels are justified by gene banks as they are described in materials and methods sections.  

  1. Figure 2: “The expression levels were the highest in the eggs and 1 day old adults.” The same is not reflected in the figure. There is a discrepancy in the figure caption and the figure.

Author’s response: Required revision has been done according to your sensible point.

  1. Figure 1 caption: N should be given either in upper case and lower case throughout the manuscript.

Author’s response: Required revision has been done according to your sensible point.

  1. Figure 1 caption: The x axis labels in the figures do not correspond or are missing in the figure caption. For eg. Gut is represented as GT in caption and G in x-axis label.

Author’s response: Required revision has been done according to your sensible point.

  1. Figure 1B and 1C mRNA expression levels do not match that with figure 1A. For instance: expression levels in guts in 1B and 1C are many folds higher than shown for the stages shown in figure 1A.

Author’s response: We appreciate your concern. Please note that the mentioned mRNA expression is related to various T. molitor life stage collection; Fig2 A different developmental Stages, 2B late instar larvae, and 2C are adults. Therefore, it is not possible to record the same expression pattern. While, performing the same experiments in three independent groups did confirm the accuracy of the relevant expressions.  

  1. Figure 2 does not align with figure 1 data. Address the comment with respect to comment 6.

Author’s response: We assume that there is a miss reference regarding comments 6 and 7. Please confirm.

  1. Section 2.3: “Given that our findings indicated……” the statement is very misleading as the Toll3 expression was not induced in all the infections (figure 2). This needs to be justified.

Author’s response: We appreciate your concern. Kindly note that Figure2 dose not present any infection condition and present developmental and tissue specific. Please note that regarding systemic infection condition, C. albicans and S. aureus induce TmToll-3 in all tested tissues and besides in the Malpighian tubules E. coli also the relevant expression.

  1. It is interesting that even after RNAi addition, there is around 0.25 fold expression observed. Has such a fold increase been recorded before?

Author’s response: As far as our understanding and knowledge about RNAi technique, you can not proceed with you knockdown experiments unless the gene of your interest has been knocked down to 75%. Thus, 75% present knockdown is considered as a common observation as it was reported and mentioned in the references about other Toll genes in T. molitor.

  1. How were the survivability percentage calculated? The values from the graph does not look to be in whole number percentages since the number of individuals considered is a whole number of 10. It will be good to mention the formula used for calculation.

Author’s response: Thank you for this fine notice. Please note that as it is mentioned in the relevant figure legend “the survival analysis was performed using Kaplan–Meier plots (log-rank chi-squared test; *p < 0.05)”

  1. The discussion is very short and does not discuss the results. The authors need to explain how the results are related between sections and with other data.

Author’s response: Thank you for another fine notice. Please note that this research is being presented in 6 graphs. Accordingly, all related conclusions and hypothesis has been discussed. We would appreciate it if you be more specific you find any data has been left undescribed.

  1. Page 8: Line 242: “their possible ligands, spzs isoforms….” This is not shown as a results in the study?

Author’s response: Thank you for another fine notice. Please note that relevant results have been reported by our team and due to your sensible point their references have been added to the manuscript.

  1. Page 8: line 256: “cuticle formation during insect metamorphosis..” mention that the results were observed for Drosophila and not T. molitor.

Authors Response: We appreciate your concern regarding the referred issue. Kindly note that it has been modified according to your request.

  1. Table 1: mention the annealing temperatures for each primer used for the qpcr study.

Authors Response: Kindly note that annealing temperature of the primers mentioned is Table 1 has been mentioned in the material and method section (Line 346) “an initial denaturation step at 94°C for 2 min followed by 35 cycles of denaturation at 94°C for 30 s, annealing at 53°C for 30 s, and extension at 72°C for 30 s, with a final extension step at 72°C for 5 min” and adding this particular information in the Table of primer sequence information shall be considered as irrelevant.

  1. Table 1: mention symbol mentioned at the end of the table at the appropriate primer names.

Author’s Response: Kindly note that it has been modified according to your request.

  1. Figure 5 can be shown as a single heatmap rather than representing it in so many graphs.

Author’s Response: Kindly be informed that the requested heatmap shall be added as supplementary.

  1. Page 2: Line 81: “Several studies…” cite reference.

Author’s response: Thank you for this fine notice. Please be informed that this sentence is the closing sentence of the last paragraph of introduction section and related references has been mentioned in the detailed sentences.

  1. There are few typographical errors and scientific misrepresentations in the manuscript. This needs to be proof-read and corrected.

Author’s response: Thank you for this fine notice. Required revision has been done according to your sensible points.

  1. Since the study is completely based on qpcr results, it will be good and interesting to show the significant results of the study in terms of protein expression. A western blot with fold change protein expression can be given for the major results to prove the hypothesis in the study.

Author’s Response: We appreciate your concern regarding the referred issue. However, due to the fact that related antibodies are not commercially available, our team is trying to purify the proteins of interest. This, however, is our future direction of innate immunity studies in T. molitor.

  1. Provide all qpcr generated graphs as a supplementary file.

Author’s response: Thank you for all your fine critics on this manuscript including the above matter. Kindly be informed that we are willing to send you the raw excel file extracted from qPCR software. Please confirm if our understanding from your inquiry is correct.

Reviewer 2 Report

Comments and Suggestions for Authors

The aim of the article by Maryam Ali Mohammadie Kojour et al. was to reveal the role of the Toll-like receptor Tm Toll-3 of Tenebrio molitor in vivo after infection with Tenebrio molitor. The mechanism of action was tentatively explained by gene knockdown and other experiments, and may be related to the indirect induction of AMP (antimicrobial peptide) expression. The structure of the article is clear but there is still room for improvement, and the following are suggestions for the article.

1. The description of Toll-3 receptor should be added to the Introduction section of the article, such as previous studies on this receptor, etc. The main object of this article is Toll-3, but only a short description of it appears in line 32, while the description of the rest of Toll receptors occupies most of the space, so it is suggested that the Introduction section should be revised.

2. Line 156, Figure 3, the red baseline of PBS is suggested to be extended a bit.

3, Line 218, Figure 5, there are several overlapping figures on the bar graphs, such as Figure 5B, the labeling of figures above group Ca is not clear, and the same is true for Figures 5C, D, F, I, J, K, M.

Author Response

Rebuttal letter to reviewer’s comments on our submitted manuscript (2294615)

We received the reviewers’ comments for the submitted manuscript (2294615). We are sincerely grateful for the reviewers’ positive and constructive comments, which have been very helpful in improving the quality of our paper. The suggestions have been noted, and we have worked on the reviewer comments. We have taken care of all comments and manuscript has been edited based on the comments. We hope the manuscript has improved with the suggested revision and would be a novel statement to the scientific community covered by the journal.

Please find the author’s comments to reviewers queries as under:

Independent Review Report:

Reviewer #2 (Comments to the Author): 

The aim of the article by Maryam Ali Mohammadie Kojour et al. was to reveal the role of the Toll-like receptor Tm Toll-3 of Tenebrio molitor in vivo after infection with Tenebrio molitor. The mechanism of action was tentatively explained by gene knockdown and other experiments, and may be related to the indirect induction of AMP (antimicrobial peptide) expression. The structure of the article is clear but there is still room for improvement, and the following are suggestions for the article.

  1. The description of Toll-3 receptor should be added to the Introduction section of the article, such as previous studies on this receptor, etc. The main object of this article is Toll-3, but only a short description of it appears in line 32, while the description of the rest of Toll receptors occupies most of the space, so it is suggested that the Introduction section should be revised.

Author’s response: We are deeply grateful for your constructive and fine comments. Kindly mark that seven TLR have been identified in T. molitor and TmToll-3 in one the isoforms. In this study our preliminary research illustrated TmToll-3 function for the very first time. Therefore, due to the lack of information about TmToll-3, the introduction section contains reports on other Toll isoforms mainly.

  1. Line 156, Figure 3, the red baseline of PBS is suggested to be extended a bit.

Author’s response: Thank you for this fine notice. Required revision has been done according to your sensible points.

3, Line 218, Figure 5, there are several overlapping figures on the bar graphs, such as Figure 5B, the labeling of figures above group Ca is not clear, and the same is true for Figures 5C, D, F, I, J, K, M.

Author’s response: Thank you for all your fine critics on this manuscript including the above matter. Required revision has been done according to your sensible points.

Author Response

Rebuttal letter to reviewer’s comments on our submitted manuscript (2294615)

We received the reviewers’ comments for the submitted manuscript (2294615). We are sincerely grateful for the reviewers’ positive and constructive comments, which have been very helpful in improving the quality of our paper. The suggestions have been noted, and we have worked on the reviewer comments. We have taken care of all comments and manuscript has been edited based on the comments. We hope the manuscript has improved with the suggested revision and would be a novel statement to the scientific community covered by the journal.

Please find the author’s comments to reviewers queries as under:

Independent Review Report:

Reviewer #3 (Comments to the Author): 

Toll like receptor (TLRs) are major players of innate immune response to pathogens in mammals and insects. However, most of these TLRs have been functionally characterized only in mammals or Drosophila and their role in innate immunity remains obscure in other insects. In this article, the authors aimed to characterize TLR Tm-Toll 3 in yellow mealworm beetle (Tenebrio molitor). According to the data presented in the paper, Tm-Toll 3 represses the expression of many antimicrobial peptides (AMPs) that lead to higher susceptibility of T. molitor to E. coli infection. Additionally, Tm-Toll 3 functions via Toll- signaling pathway to regulate AMP expression. I would like to suggest following revisions:

  1. Lines 69-71: “Unlike mammalian TLRs, T. molitor TLRs 69 are not directly triggered by structurally conserved molecules derived from microbes or 70 other possible stimuli but are instead activated by the cytokine ligand Spätzle (Spz)” Reference is missing for this information.

Author’s response: We are deeply grateful for your constructive and fine comments. Kindly mark that the relevant reference has been added to the text according to your request.

  1. Line 78-79: “Tenebrio modular serine protease (ModSP) activates a cascade of CLIP-domain

zymogens 78 comprising the serine protease Spätzle-processing enzyme activating enzyme (SAE)

and SPE” SPE should come in parentheses after Spätzle-processing enzyme.

Author’s response: Thank you for another fine notice. Kindly mark that the mentioned sentence has been revised according to your sensible request.

  1. Figure 2 legend says Tm-Toll3 expression in highest in eggs and 1 day old adults, however

according to the graph, the expression level is not so high in 1 day old adult, but is high in 5 and

6 year old pupae (P5 and P6). Is this a typing error? If not, please explain in more detail.

Author’s response: Thank you for another fine notice. Kindly mark that the mentioned sentence was a typing error, and it has been revised accordingly.

  1. In Figure 3A, there is an extra asterisk before 6H reading. Additionally, the figure legend says that highest TmToll-3 161 expression level was observed in the hemolymph and Malpighian tubules in response to C. albicans 162 challenge. However, according to figures 3B and 3D, there is higher expression in the gut and fat bodies as compared to the hemolymph. Please rectify the legend.

Author’s response: Thank you for all your fine critics on this manuscript including the above matter. Please note that the mentioned mistakes have been revised according to your sensible request.

Reviewer 4 Report

The manuscript is precise and elaborative, while summarizing the role of the Tm Toll-3 receptor in immune response using T.molitor as model organism. I want to thank the authors for selecting this topic. This paper is very much relevant with the journal's interest. Although the paper contains adequate data but there are few areas in this paper which needs more explanation. Therefore, I recommend a minor revision for the manuscript. I explain my concerns in more detail below. I ask that the authors to specifically address each of my comments in their response.

 Minor Comments:

1. T.molitor in not harmful to humans as far my knowledge. And roll of different TLRs have already been extensively studied in Drosophila. So, the authors need to clarify the importance of using T.molitor as another model insect organism more clearly in the introduction section.

2. In Figure 3, the Y axis value for sub figures A-D are different. The authors should use a standardized value range for all sub figures, so that the mRNA expression of TmToll-3 receptors can be compared in various aprts of the body against PBS control.

3. In figure 4D, the control dsRNA (TmVer) is missing as tere is only red line.

4. How the infections were given to T.molitor need to be clearly mentioned in the materials and methods section.

5.The same Table 1 has been mentioned as supplementary table 1 in the manuscript. Please clarify whether they are same table or not?

Author Response

Rebuttal letter to reviewer’s comments on our submitted manuscript (2294615)

We received the reviewers’ comments for the submitted manuscript (2294615). We are sincerely grateful for the reviewers’ positive and constructive comments, which have been very helpful in improving the quality of our paper. The suggestions have been noted, and we have worked on the reviewer comments. We have taken care of all comments and manuscript has been edited based on the comments. We hope the manuscript has improved with the suggested revision and would be a novel statement to the scientific community covered by the journal.

Please find the author’s comments to reviewers queries as under:

Independent Review Report:

Reviewer #4 (Comments to the Author): 

 The manuscript is precise and elaborative, while summarizing the role of the Tm Toll-3 receptor in immune response using T.molitor as model organism. I want to thank the authors for selecting this topic. This paper is very much relevant with the journal's interest. Although the paper contains adequate data but there are few areas in this paper which needs more explanation. Therefore, I recommend a minor revision for the manuscript. I explain my concerns in more detail below. I ask that the authors to specifically address each of my comments in their response.

 Minor Comments:

  1. molitor in not harmful to humans as far my knowledge. And roll of different TLRs have already been extensively studied in Drosophila. So, the authors need to clarify the importance of using T.molitor as another model insect organism more clearly in the introduction section.

Author’s response: We are deeply grateful for your constructive and fine comments. kindly notice that despite importance of Drosophila as a powerful study model and as essential tool to study mechanisms underlying numerous human genetic diseases, it is still limited in terms of determining the biochemical mechanisms involved in regulating the proteolytic Toll cascade (extracellular part of activation chain). Since Drosophila has several alternative routes to the Toll pathway, used at various developmental stages and infection protocol, it is difficult to determine the clear activation mechanism. For instance, identification of downstream factors of Persephone is yet to be clear. Moreover, we also mentioned T. molitor particular importance in so many industries and their nutritional values have been mentioned in last paragraph of the introduction. However, since this is not a review literature, we avoid further explanation.

  1. In Figure 3, the Y axis value for sub figures A-D are different. The authors should use a standardized value range for all sub figures, so that the mRNA expression of TmToll-3 receptors can be compared in various aprts of the body against PBS control.

Author’s response: Thank you for another fine notice. Kindly mark that we have used different value range so that the readers not only can compare the TmToll-3 expression within all dissected tissues, but also they can clearly compare the expression following the infection in different time points.

  1. In figure 4D, the control dsRNA (TmVer) is missing as tere is only red line.

Author’s response: Thank you for another fine notice. Kindly note that the mentioned expression is not missing but has been overlapped by dsTmToll-3 expression survivability.

  1. How the infections were given to T.molitor need to be clearly mentioned in the materials and methods section.

Authors Response: We appreciate your concern regarding the referred issue. Kindly note that microbial concentration of each microbial source, culture condition, and systemically infected groups have be mentioned clearly in 4.1. section of materials and methods. However, we would appreciate it if mention your required data in more detail.

5.The same Table 1 has been mentioned as supplementary table 1 in the manuscript. Please clarify whether they are same table or not?

Author’s response: Thank you for all your fine critics on this manuscript including the above matter. Please note that the mentioned mistakes have been revised according to your sensible request.

Round 2

Reviewer 1 Report

The authors have addressed the comments satisfactorily. However, I would like the author's comment on the following issues. 

1. The annealing temperature is usually different for different primers. However, in the study, you have used a single temperature. This might lead to some primer binding to the template and the other not binding or binding weakly. This could lead to a significant error (lower amplification or non-specific amplification in the qpcr results. The authors should provide a reason for using a single temperature here. 

2. Provide a data availability statement for the qpcr raw files at the end. 

3. Supplementary figure 2 is not cited in the manuscript. 

Author Response

We hope the manuscript has improved with the suggested revision and would be a novel statement to the scientific community covered by the journal.

Please find the author’s comments to reviewers queries as under:

Independent Review Report (Second round):

Reviewer #1 (Comments to the Author): 

The authors have addressed the comments satisfactorily. However, I would like the author's comment on the following issues. 

  1. The annealing temperature is usually different for different primers. However, in the study, you have used a single temperature. This might lead to some primer binding to the template and the other not binding or binding weakly. This could lead to a significant error (lower amplification or non-specific amplification in the qpcr results. The authors should provide a reason for using a single temperature here.

Author’s response: We appreciate your concern. Please note that primer annealing temperature have been adjusted during the primer design similar to GC content and primer length.

  1. Provide a data availability statement for the qpcr raw files at the end. 

Author’s response: We have the data availability statement at the end of manuscript and relevant supplementary data have been have been added to the manuscript materials.

  1. Supplementary figure 2 is not cited in the manuscript. 

Author’s response: Kindly note that mentioned supplementary data has been cited in line 203.

Sincerely,

Yeon Soo Han,

(Corresponding author)

Email: hanys@jnu.ac.kr
